# Modeling the Growth of Six *Listeria monocytogenes* Strains in Smoked Salmon Pâté

**DOI:** 10.3390/foods12061123

**Published:** 2023-03-07

**Authors:** Araceli Bolívar, Chajira Garrote Achou, Fatih Tarlak, María Jesús Cantalejo, Jean Carlos Correia Peres Costa, Fernando Pérez-Rodríguez

**Affiliations:** 1Department of Food Science and Technology, UIC Zoonosis y Enfermedades Emergentes ENZOEM, ceiA3, Universidad de Córdoba, 14014 Córdoba, Spain; 2Department of Agronomy, Biotechnology and Food, School of Agriculture Engineering, Public University of Navarre (UPNA), Campus de Arrosadia, 31006 Pamplona, Spain; 3Department of Nutrition and Dietetics, Istanbul Gedik University, 34876 Istanbul, Turkey

**Keywords:** predictive microbiology, Ratkowsky secondary model, shelf life, foodborne pathogen, ready-to-eat fish product

## Abstract

In this study, the growth of six *L. monocytogenes* strains isolated from different fish products was quantified and modeled in smoked salmon pâté at a temperature ranging from 2 to 20 °C. The experimental data obtained for each strain was fitted to the primary growth model of Baranyi and Roberts to estimate the following kinetic parameters: lag phase (*λ*), maximum specific growth rate (*μ_max_*), and maximum cell density (*N_max_*). Then, the effect of storage temperature on the obtained *μ_max_* values was modeled by the Ratkowsky secondary model. In general, the six *L. monocytogenes* strains showed rapid growth in salmon pâté at all storage temperatures, with a relatively short lag phase λ, even at 2 °C. The growth behavior among the tested strains was similar at the same storage temperature, although significant differences were found for the parameters *λ* and *μ_max_*. Besides, the growth variations among the strains did not follow a regular pattern. The estimated secondary model parameter *T_min_* ranged from −4.25 to −3.19 °C. This study provides accurate predictive models for the growth of *L. monocytogenes* in fish pâtés that can be used in shelf life and microbial risk assessment studies. In addition, the models generated in this work can be implemented in predictive modeling tools and repositories that can be reliably and easily used by the fish industry and end-users to establish measures aimed at controlling the growth of *L. monocytogenes* in fish-based pâtés.

## 1. Introduction

In recent years, there has been a significant increase in the development of new fish-based foods that are easy to prepare and consume. Among these, fish-based food emulsions, such as nuggets, hamburgers, pâtés, and sausages, stand out due to their consolidation in the market and their great acceptance among consumers [1]. Pâté is a spreadable emulsion-type food product with an important gastronomic tradition and highly appreciated sensory properties [2]. Moreover, there has been a diversification with different sources other than meat being used for pâté making, such as fish [3]. In general, fish species with significant commercial value (e.g., salmon, tuna, or anchovy) are used for the production of fish-based pâtés [4,5]. Smoked salmon is one of the most common choices for fish pâté production, especially in Northern European countries. 

Wide varieties of pâtés are commercialized as ready-to-eat (RTE) delicatessen products since no reheating or further cooking is required before consumption. Depending on the heat treatment applied during processing (usually pasteurization or sterilization), pâtés produced by food companies can have a relatively long shelf life, varying from ca. 1 month for pasteurized products requiring refrigerated storage up to 24 months for sterilized products stored at room temperature. Although pâté products are usually heat-treated, microbial recontamination can occur either before packaging or later at the consumer stage.

Recent data have highlighted that *Listeria monocytogenes*, the causative agent of human listeriosis, remains a primary concern for the food industry, especially in the cold chain of RTE food products. In this regard, 86 alert notifications have been recorded by the European Rapid Alert System for Food and Feed (RASFF) from 2019 to 2023 linked to the presence of *L. monocytogenes* in fish products, mainly smoked salmon. This represents 75% of the alert notifications related to the presence of pathogenic microorganisms in fish products in the last five years in the European Union (EU). Although listeriosis is the fifth most reported zoonosis in humans in the EU, it represents the highest proportion of hospitalized cases of all zoonoses under EU monitoring, with a case fatality rate of 13.7% [6]. 

In a wide baseline survey conducted by the European Food Safety Authority (EFSA) during 2010–2011 on the prevalence of *L. monocytogenes* in RTE foods at retail, the highest prevalence levels were found in RTE fish products (10.3%) compared to RTE meat (2.07%) and cheese products (0.47%) at the end of shelf life [7]. In the latter study, a quantitative risk assessment of *L. monocytogenes* was carried out in the abovementioned food categories, showing that pâté presented the highest listeriosis risk estimates within the category of heat-treated meat products [8]. In a survey on the prevalence and contamination levels of *L. monocytogenes* in pâtés sold in Spain, the pathogen was detected in 5.4% of samples (n = 182), with one sample containing more than 10^2^ log CFU/g [9]. Various listeriosis outbreaks have involved pâté products [10,11]. Recently, *L. monocytogenes* has been detected in smoked salmon pâté products commercialized by food producing companies from Ireland [12] and Australia [13]. These events demonstrate the challenges associated with the control of *L. monocytogenes* in this food category because its intrinsic factors (e.g., pH > 5.0, a_w_ > 0.94) and refrigerated storage allow for pathogen growth. 

Predictive microbiology is a useful approach to describing food systems and predicting the behavior of microorganisms in foods through mathematical equations and algorithms [14]. Recent modeling studies have shown that food microstructure has a significant influence on microbial growth dynamics, but its effect is complex and seems to be dependent on storage temperatures and the specific compositional and rheological properties of the target food matrix [15,16]. One of the most interesting findings of the mentioned studies is that the presence of fat droplets enhanced the growth of *L. monocytogenes* at low storage temperatures (4 °C) in fish-based emulsions, which suggests that fat content may have an impact on the safety of fat-containing RTE foods.

Food-oriented approaches based on challenge testing are recommended for the development of more accurate predictive microbiology models. In addition, using pathogen strains isolated from particular foods would result in more accurate model predictions since those strains would be better adapted to the microecological conditions of the target food as compared, for instance, to clinical isolates. Despite growth variations among strains, predictive models have been traditionally generated using only one or a small cocktail of strains. In this regard, different authors have illustrated the importance of studying strains from different origins to assess strain variability when developing predictive growth models [17,18].

Studies that have quantified and modeled the growth of *L. monocytogenes* in pâtés other than liver-based pâtés are scarce in the literature [19,20,21]. Other pâté-type products, such as those made from fish, are relevant due to their widespread consumption and added value. Therefore, the objective of this study was to quantify and model the growth of six *L. monocytogenes* strains isolated from different fish products in smoked salmon pâté in a wide temperature range from 2 to 20 °C.

## 2. Materials and Methods

### 2.1. Smoked Salmon Pâté 

A commercial, heat-treated (sterilized) smoked salmon pâté was acquired from a Spanish food company. The pâté was received in single-dose containers (23 g) made of semi-rigid aluminum foil and packaged under a normal (aerobic) atmosphere. The composition was smoked salmon (29%), sunflower oil, water, milk, potato, carrageenan, salt (1%), and paprika extract.

### 2.2. Bacterial Strains and Culture Conditions

Six *L. monocytogenes* strains isolated from different fish products were selected in this study. The strains LMG 23773 (serotype 4) and LMG 23774 (serotype 1), which were both isolated from smoked salmon, and LMG 26484 (serotype not available), isolated from tuna salad, were acquired from the BCCM/LMG bacterial collection of Ghent University (Belgium). The strains 12MOB101LM (serotype 1/2b), 12MOB102LM (serotype 4b), and 12MOB107LM (serotype 4b), which were isolated from hareng, salmon, and trout, respectively, were acquired from the set of *L. monocytogenes* strains for conducting challenge tests at the EU Reference Laboratory for *L. monocytogenes* (EURL Lm) [22].

The stock cultures were stored in 20% glycerol at −80 °C. The strains were separately cultured by transferring a loopful into 9 mL of Brain Heart Infusion broth (BHI; Oxoid, Basingstoke, UK) and incubating at 30 °C for 24 h. Next, two consecutive subcultures were prepared at the same conditions, with 18-20 h of incubation for the latter subculture. The cultures were washed twice in sterile saline solution (SS; 0.85% *w/v* NaCl) by centrifugation at 3326× *g* (5810R; Eppendorf, Hamburg, Germany) for 10 min and then 10-fold diluted in SS to obtain a target inoculum of 10^2^ CFU/g in the salmon pâté.

### 2.3. Challenge Tests

The single-dose containers of salmon pâté (which were considered the analytical samples) were individually inoculated with each *L. monocytogenes* strain at ca. 10^2^ CFU/g. This contamination level was set according to the EURL Lm Technical Guidance Document in order to avoid measuring uncertainty associated with low numbers when using enumeration methods [23]. Prior to inoculation, an adhesive septum (ø 15 mm; PBI Dansensor, Ringsted, Denmark) was placed on the upper central part of the containers’ aluminum foil lid. Then, aliquots of 50 µL (2% *v*/*w*) were aseptically inoculated in the pâté containers (not on the product surface) using a sterile syringe with a needle (BD Plastipak, Madrid, Spain) inserted through the adhesive septum [24]. A non-inoculated (control) batch made up of four pâté samples was prepared for each experimental condition. The inoculated and non-inoculated samples were stored at constant temperatures representative for the storage of fish products, i.e., 2 °C (cold storage), 8 °C (domestic refrigeration), and 14 and 20 °C (mild abuse temperatures). For each strain and temperature, the experiments were independently carried out in triplicate.

### 2.4. Microbiological Analyses

At each sampling point, two pâté samples were analyzed for *L. monocytogenes* enumeration according to ISO 11290-2:2017. To this end, the samples were homogenized with 0.1% peptone water (Oxoid, UK) for 1 min using a stomacher (IUL Instruments, Barcelona, Spain). The homogenates were surface-plated on Listeria selective agar (Oxford formulation; Oxoid, UK) with a selective supplement (SR140E; Oxoid, UK), and the plates were incubated at 37 °C for 48 h. In addition, the total aerobic mesophilic bacteria counts were determined in the non-inoculated samples at the beginning and end of the experiments by plating the homogenates in plate count agar (PCA; Oxoid, UK) and incubating for 48–72 h at 30 °C.

### 2.5. Physicochemical Analyses

The physicochemical properties (pH and a_w_) of the non-inoculated pâté samples were determined in triplicate. A pH meter (Edge HI2020; Hanna Instruments, Woonsocket, RI, USA) connected to an electrode for semi-solids and emulsions (HI10530; Hanna Instruments, USA) was used to determine the pH. The a_w_ was measured using AquaLab (Series 4TE; Decagon Devices Inc., Pullman, WA, USA).

### 2.6. Primary Model Fitting

The explicit form of the primary growth model of Baranyi and Roberts [25], represented by Equations (1)–(3), was fitted to the growth curves obtained for each *L. monocytogenes* strain at the studied storage temperatures. This model was used to obtain the kinetic growth parameters of lag time (*λ*; h), maximum specific growth rate (*μ_max_*; 1/h), and maximum population density (*N_max_*; ln CFU/g). The model parameters were estimated from the set of experimental data via the minimization of the sum of squared errors, using the lsqnonlin routine of the optimization toolbox of Matlab, version R2022a (The Mathworks Inc., Portola Valley, CA, USA). The standard errors of the parameter estimates were calculated from the Jacobian matrix obtained in the fitting process.
(1)$$N\left(t\right)=N\left(0\right)+\mu_{max}\cdot A\left(t\right)-ln\left(1+\frac{e^{\mu_{max}\cdot A\left(t\right)}-1}{e^{\left(N_{max}-N\left(0\right)\right)}}\right)$$
(2)$$A\left(t\right)=t+\frac{1}{\mu_{max}}ln\left(\frac{e^{-\mu_{max}\cdot t}+Q\left(0\right)}{1+Q\left(0\right)}\right)$$
(3)$$Q\left(0\right)=\frac{1}{e^{\mu_{max}\cdot \lambda }-1}$$
where *N*(*t*) [ln CFU/g] is the cell density at time *t*; *N_0_* [ln CFU/g] is the initial cell density at time 0; *N_max_* [ln CFU/g] is the maximum cell density; *μ_max_* [1/h] is the maximum specific growth rate; *Q*[−] is a dimensionless parameter characterizing the physiological state of the cells; *Q*(0) is a measure of the initial physiological state of the cells; *λ* [h] is the lag time of the cells.

### 2.7. Secondary Model Fitting

The maximum growth rate (*μ_max_*) values estimated by the primary model were used to develop the square-root-type model of Ratkowsky et al. [26] (Equation (4)). This model was used to describe the effect of storage temperature on *μ_max_*.
(4)$$\sqrt{\mu_{max}}=b\left(T-T_{min}\right)$$
where *b* is a constant; *T* (°C) is the storage temperature; *T_min_* (°C) is the theoretical minimum growth temperature. The NonLinearModel.fit command was used in the Matlab software to obtain the secondary model parameters [27].

### 2.8. Statistical Analysis

The growth kinetic parameters estimated by the primary model (i.e., *λ* and *μ_max_*) and the obtained parameters of the secondary model (i.e., *b* and *T_min_*) were statistically compared for the different strains by a one-way analysis of variance (ANOVA) using the Minitab statistical software. The Shapiro–Wilk test and Levene’s test were previously applied to check the normality and homogeneity of the data. The statistical differences between the strains were determined by post hoc analysis using Tukey’s multiple range test with a *p* value of ≤0.05. 

## 3. Results

The growth of the studied *L. monocytogenes* strains at 2, 8, 14, and 20 °C in smoked salmon pâté is illustrated in Figure 1A, Figure 1B, Figure 1C, and Figure 1D, respectively. All of the strains grew at the different studied temperatures. The three growth curve phases can be observed at the tested temperatures for the different *L. monocytogenes* strains, although the lag phase was not observed for all strains at temperatures of >8 °C. The stationary phase was reached after approximately 1000, 250, 100, and 50 h at 2, 8, 14, and 20 °C, respectively. As expected, the studied storage temperatures did not have a significant influence on the maximum cell density *N_max_* of the pathogen strains in the fish pâté, with an average of 20.2 ± 0.4, 20.1 ± 0.4, 20.3 ± 0.2, and 20.3 ± 0.3 ln CFU/g at 2, 8, 14, and 20 °C, respectively. Hence, only the kinetic parameters *λ* and *μ_max_* will be referred to in the results and related discussion presented hereafter.

The estimated *λ* and *μ_max_* values of the six *L. monocytogenes* strains in the salmon pâté at 2–20 °C by the fitted Baranyi model are provided in Table 1. The statistical analysis revealed that there were no significant differences (*p* > 0.05) in the *λ* among the *L. monocytogenes* strains for the same storage temperatures, except at 2 °C, where the *λ* of the strain 12MOB102LM was significantly shorter (*p* ≤ 0.05) and even not observed at 8 °C. At the highest tested temperature (20 °C), most of the strains did not present *λ*, although a very short *λ* (<2 h) was observed for the LMG 26484 and 12MOB107LM strains. On the contrary, significant differences in *μ_max_* (*p* ≤ 0.05) were found between the strains for the same storage temperatures, especially at 20 °C, with the *μ_max_* values ranging from 0.261 to 0.330 1/h. The slowest growth was observed at 2 °C for the LMG 23774 strain, with *λ* and *μ_max_* values of 102 h and 0.017 1/h, respectively. However, the growth variations among the strains did not follow a regular pattern. Additionally, the growth behavior of the three *L. monocytogenes* strains acquired from the Belgian collection was quite similar to the three strains from the EURL Lm, as observed in Figure 1 and Table 1.

Secondary models were developed using the predicted *μ_max_* values in salmon pâté for each *L. monocytogenes* strain. The graphical version of the fitted models is illustrated in Figure 2. The parameters of the developed models are shown in Table 2. As it can be observed, the relationship between √*μ_max_* and the storage temperature was linear, with high R^2^ values ranging from 0.982 to 0.999. This indicates that the temperature dependence was properly described by the square-root-type model. In addition, the obtained model parameters *b* and *T_min_* were not significantly different (*p* > 0.05) among the pathogen strains, with very similar *b* values (0.021–0.024). The lowest *T_min_* was obtained for the LMG 23773 strain, with a value of −4.25 °C. 

## 4. Discussion

The physicochemical characteristics of the studied salmon pâté corresponded to a pH of 6.05 ± 0.01 and a_w_ of 0.986 ± 0.003. These values are in line with those reported by other authors for fish-based pâtés [28,29,30]. The total bacterial count in the non-inoculated pâté samples remained below the limit of quantification (<10 CFU/g) during the experiments, probably due to the sterilization applied during the processing of the studied fish pâté. 

In general, the six *L. monocytogenes* strains showed rapid growth in the salmon pâté at all storage temperatures, with a relatively short lag phase *λ*, even at 2 °C. This could be explained by the compositional and intrinsic properties of the tested fish pâté, together with the absence of background microbiota, which was removed as a consequence of the high heat treatment process applied during manufacturing. The storage temperatures clearly affected the pathogen’s growth, with the lowest growth kinetics at 2 °C. The effect of temperature on microbial growth dynamics has been largely described in the past [26]. The growth behavior among the strains was similar at the same storage temperatures, although significant differences were found for the parameters *λ* and *μ_max_*. The variability in *μ_max_* among the strains increased with the temperature increase. This result may differ with findings from previous works, which suggested that the strain variability among *L. monocytogenes* strains was greater when the growth conditions became unfavorable [17,31,32]. However, these studies have mostly been performed in non-selective broth media. Therefore, our knowledge of the variations among *L. monocytogenes* isolates based on growth in the studied fish product is largely limited.

In the present work, the growth variation among the strains did not follow a regular trend, which is in line with other *L. monocytogenes* growth comparative studies [33]. Besides, a similar growth behavior was observed between the set of strains from the Belgian collection and the set of reference strains from the EURL Lm. In this regard, wild-type strains are more likely to adapt and grow on the food matrix as compared to reference strains [34]. The findings of this study could be explained by the fact that the six tested strains are isolates from fish products, which might enhance their adaptation to the particular characteristics of the studied fish pâté.

The growth dynamics of *L. monocytogenes* have been barely investigated in fish-based pâtés. Farber and Daley [28] assessed the ability of an *L. monocytogenes* meat isolate to grow on salmon pâté over a 3-week storage period at 4 °C. They observed that the pathogen was able to survive but not grow after 21 days of storage. This may be related to the strain and/or compositional factors of the fish pâté used in the experiments. Nielsen et al. [30] studied the growth of a two-strain mixture of *L. monocytogenes* (strain CCUG 32,964 and strain SIK609, isolated from fish) in salmon pâté during storage at 8 °C, reporting a value for *μ_max_* of 0.0768 1/h. This value was almost equal to the highest *μ_max_* obtained at 8 °C in our study (0.0761 1/h), which was observed for the LMG 26484 strain. Verheyen et al. [15] investigated the growth dynamics of *L. monocytogenes* in fish-based gelled emulsion systems using a cocktail of the same strains as those used in the present work from the Belgian collection (LMG 23773, LMG 23774, and LMG 26484). The experimental model systems were developed with a fat content ranging from 1 to 20% in order to mimic the compositional and physicochemical properties of real processed fish products (e.g., surimi, fish sausage, and fish pâté). The findings reported by Verheyen et al. [15] in gelled emulsions with 20% fat obtained at 7 °C (which were the closest experimental conditions to those used in our work) were compared to the data from the current study conducted in salmon pâté. The pathogen’s lag phase *λ* value in the gelled emulsions with 20% fat at 7 °C was 25.13 ± 4.19 h, which was slightly longer than the *λ* values estimated for the different strains in salmon pâté at 8 °C (6.60–22.62 h), probably due to the difference between the storage temperatures used in the experiments. Similarly, the *μ_max_* in the gelled emulsions (0.0578 1/h) was slightly lower than the *μ_max_* values observed in the studied fish pâté at 8 °C, although it was almost equal to the *μ_max_* obtained for the LMG 23773 and 12MOB102LM strains, i.e., 0.0589 and 0.0591 1/h, respectively. Interestingly, these observations suggest that the difference in fat content between the gelled emulsions (20%) and the studied fish pâté (30%) does not affect the pathogen’s *μ_max_*. The ability of *L. monocytogenes* to grow in foods containing a relatively high fat content has been described in the literature and could be explained by the confocal laser scanning microscopy images obtained by Verheyen et al. [15] in the gelled emulsions with 20% fat, which demonstrated that the growth of *L. monocytogenes* occurred mostly as microcolonies around the fat droplets, that is, on the fat–water interface. It should be noted, however, that the affinity of the cells to grow on the fat–water interface might be dependent on the specific pathogen strain and growth temperature.

Findings obtained by other authors in meat-based pâtes were compared to the results of the current study, although to a lesser extent due to the use of different storage temperatures and product formulations. For instance, Farber et al. [20] modeled the effect of various factors, including storage temperature (4 and 10 °C), NaCl, sodium nitrite, sodium erythrobate, and spice, on the growth of *L. monocytogenes* in pork liver pâtés. The pathogen grew well in all experimental conditions, with an increase in cell density varying from 6.08 to 7.04 log CFU/g. Additionally, temperature was the only factor affecting the pathogen’s *μ_max_*. Hunt et al. [35] determined that the growth potential of *L. monocytogenes* in pork liver pâté at 8 °C during 7 days of storage was around 3 log, but they did not report information about growth kinetics. The ability of the pathogen to grow in meat pâtés was also described by Hayrapetyan et al. [36], who observed that the *N_max_* (~9 log CFU/g) was reached after ca. 18, 11, and 5 days at 4, 7, and 12 °C, respectively, in commercial pâté samples.

A number of studies have evaluated the effect of storage temperature on the growth kinetics of *L. monocytogenes* in fish products, but to the best of the authors’ knowledge, data obtained in pâté-type products are not available in the literature. The estimated secondary model parameters in this work, which did not significantly differ among the strains, are in agreement with values reported by other authors for fish products. Bolívar et al. [37] developed a secondary model for the *L. monocytogenes* strain NCTC 11994 in sterile sea bream juice, reporting *T_min_* values of −3.83 and −4.16 °C under reduced oxygen and aerobic atmosphere, respectively. Similarly, Costa et al. [38] estimated a *T_min_* of −3.40 °C for the *L. monocytogenes* strain CTC1034 in the abovementioned fish-based juice. Delignette-Muller et al. [39] estimated a *T_min_* value of −2.86 °C in cold-smoked salmon for a wide set of *L. monocytogenes* strains, some of them isolated from cold-smoked salmon. A similar *T_min_* value (−2.30 °C) was estimated by Mejlholm and Dalgaard [40] in cold-smoked salmon for a mixture of four *L. monocytogenes* isolates from seafood. In the latter two studies, the background microbiota, mainly lactic acid bacteria, had an important competitive effect on the pathogen’s growth. This may explain the lower *T_min_* values estimated in our study (varying from −4.25 to −3.19 °C), reflecting the greater ability of *L. monocytogenes* to grow in the tested salmon pâté. 

In this work, the developed growth models only included the effect of storage temperature on the growth dynamics of *L. monocytogenes* in salmon pâté. Future modeling studies could also investigate the influence of other relevant factors for fish pâtés, such as compositional characteristics (e.g., fat and salt contents), atmospheric packaging conditions, and the interactive effects between them. In addition, the behavior of *L. monocytogenes* in non- or mildly heat-treated pâté-type products, in which the presence of background microbiota may have a significant effect on the pathogen’s growth, should be assessed. 

## 5. Conclusions

This study was one of the first to quantify and model the growth of *L. monocytogenes* in a commercial fish-based pâté. The results demonstrated the ability of the six tested pathogen strains to grow in the studied salmon pâté at a wide temperature range. This study provides accurate predictive models for the growth of *L. monocytogenes* in fish pâtés that can be used in shelf life and quantitative risk assessment studies. Moreover, the models generated in this work can be implemented in predictive modeling tools and repositories that can be reliably and easily used by the fish industry and end-users to establish measures aimed at controlling the growth of *L. monocytogenes* in fish-based pâtés. 

## Figures and Tables

**Figure 1 foods-12-01123-f001:**
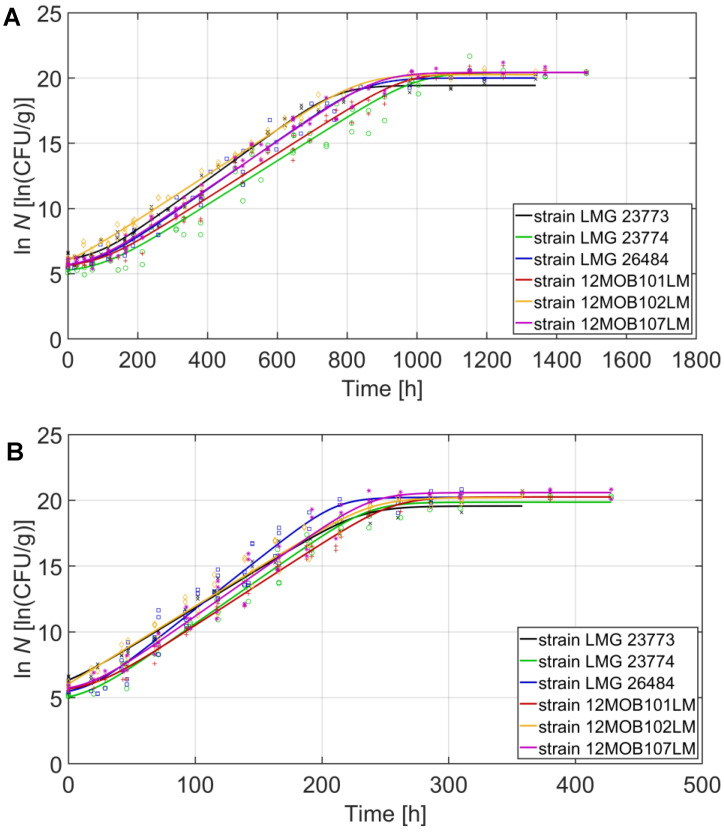
Growth of six *L. monocytogenes* strains in smoked salmon pâté at 2 (**A**), 8 (**B**), 14 (**C**), and 20 °C (**D**). The symbols (×, ○, □, +, ◊, and ✳ for strains LMG 23773, LMG 23774, LMG 26484, 12MOB101LM, 12MOB102LM, and 12MOB107LM, respectively) represent the experimental data, and the lines correspond to the primary model fit [25].

**Figure 2 foods-12-01123-f002:**
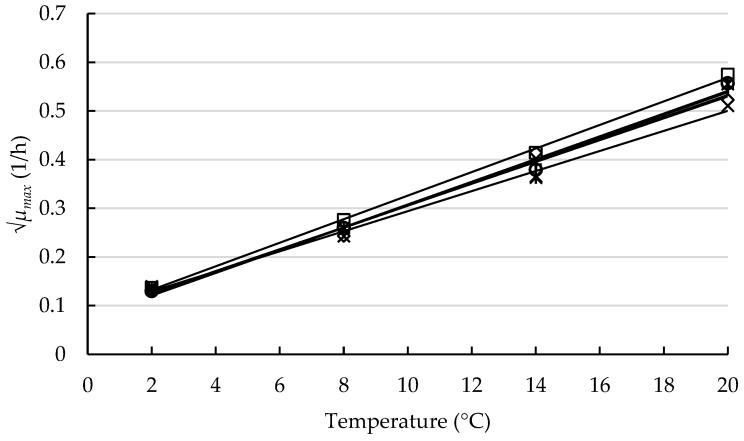
Graphical representation of the secondary models [26] developed for the six *L. monocytogenes* strains in smoked salmon pâté. The symbols (×, ○, □, +, ◊, and ✳ for strains LMG 23773, LMG 23774, LMG 26484, 12MOB101LM, 12MOB102LM, and 12MOB107LM, respectively) represent the √*μ_max_* values, and the lines correspond to the model fit.

**Table 1 foods-12-01123-t001:** Kinetic parameters ± standard errors estimated by the model of Baranyi and Roberts [25] for the growth of six *L. monocytogenes* strains in smoked salmon pâté at 2, 8, 14, and 20 °C. For the different strains at the same temperature, the values bearing different uppercase letters are significantly different (*p* ≤ 0.05).

Strain	2 °C		8 °C		14 °C		20 °C	
	*λ* (h)	*µ_max_* (1/h)	*λ* (h)	*µ_max_* (1/h)	*λ* (h)	*µ_max_* (1/h)	*λ* (h)	*µ_max_* (1/h)
LMG 23773	86.52 ± 14.68 ^A^	0.0194 ± 0.0005 ^A^	6.60 ± 7.17 ^A^	0.0589 ± 0.0022 ^C^	4.15 ± 4.34 ^A^	0.1339 ± 0.0083 ^B^	NE	0.2607 ± 0.0121 ^F^
LMG 23774	101.91 ± 29.39 ^A^	0.0168 ± 0.0006 ^C^	16.99 ± 7.03 ^A^	0.0672 ± 0.0023 ^B^	NE	0.1437 ± 0.0099 ^B^	NE	0.3098 ± 0.0178 ^B^
LMG 26484	90.58 ± 27.32 ^A^	0.0186 ± 0.0008 ^AB^	18.05 ± 7.36 ^A^	0.0761 ± 0.0035 ^A^	7.01 ± 3.07 ^A^	0.1714 ± 0.0083 ^A^	1.31 ± 2.39 ^A^	0.3300 ± 0.0239 ^A^
12MOB101LM	96.07 ± 23.07 ^A^	0.0170 ± 0.0005 ^C^	22.62 ± 5.43 ^A^	0.0625 ± 0.0016 ^BC^	NE	0.1519 ± 0.0111 ^AB^	NE	0.3002 ± 0.0196 ^D^
12MOB102LM	21.41 ± 15.47 ^B^	0.0173 ± 0.0004 ^ABC^	NE ^1^	0.0591 ± 0.0028 ^C^	8.34 ± 2.83 ^A^	0.1715 ± 0.0078 ^A^	NE	0.2733 ± 0.0104 ^E^
12MOB107LM	77.16 ± 17.12 ^AB^	0.0181 ± 0.0004 ^ABC^	19.20 ± 9.34 ^A^	0.0675 ± 0.0031 ^B^	NE	0.1316 ± 0.0081 ^B^	0.66 ± 1.30 ^A^	0.3080 ± 0.0123 ^C^

^1^ No lag estimated.

**Table 2 foods-12-01123-t002:** Parameters ± standard errors of the secondary model [26] developed for the studied *L. monocytogenes* strains in smoked salmon pâté.

Strain	*b*	*T_min_* (°C)	R^2^
LMG 23773	0.021 ± 0.001	−4.25 ± 0.73	0.995
LMG 23774	0.023 ± 0.001	−3.19 ± 1.00	0.992
LMG 26484	0.024 ± 0.001	−3.47 ± 0.41	0.999
12MOB101LM	0.023 ± 0.001	−3.20 ± 0.69	0.996
12MOB102LM	0.022 ± 0.001	−3.63 ± 1.03	0.992
12MOB107LM	0.023 ± 0.002	−3.43 ± 1.54	0.982

## Data Availability

Data are contained within the article.

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
