# Peer review of "Modeling the Growth of Six Listeria monocytogenes Strains in Smoked Salmon Pâté"

_foods, 2023, doi:10.3390/foods12061123_

Round 1
Reviewer 1 Report
This manuscript modeled the growth of six L. monocytogenes strains in smoked salmon pâté. The work provided some interesting results. However, this manuscript's quality should be improved.
1. The authors should introduce the current risk of L. monocytogenes in smoked salmon pâté products in the Introduction part, which is important to the necessity of research. Besides, I suggest that the authors provide references to the existence of L. monocytogenes in smoked salmon pâté products.
2. The growth of six strains was modeled in the smoked salmon pâté. However, none of them were isolated from the smoked salmon pâté. It confused me why the authors modeled the growth of the strains from tuna salad and others sources in the smoked salmon pâté.
3. What is the storage temperature of the smoked salmon pâté products? Why did the authors conduct the experiments at four temperatures (2, 8, 14, and 20°C)?
4. I suggest that the authors provide more academic figures.
5. What do the notes ‘-1' and ‘-' mean in Table 1?
6. In the conclusions, the authors said, ' The results demonstrated that smoked salmon pâté must be considered a high-risk food product due to its intrinsic properties and the ability of L. monocytogenes to grow in the studied product in a wide temperature range.' I do not think it is a persuasive conclusion, as the L. monocytogenes strains were added to the smoked salmon pâté.
Reviewer 2 Report
1. Line: 89-95: Do you have experimental data for each listeria inoculated below 102 CFU/g? I think it's more appropriate to include a lag phase of enough time. Since bacterial infection is an important issue when only one bacterium is divided by binary fission, and enters the stage where infection is possible. It's a very important issue to see how fast it goes up to the growth stage when there's even one bacterium. However, the data presented seem not qualified; they are too preliminary without solid analysis
Needs to be more detail (experimental data supporting the statement) and also needs references.
2. In order to apply the current study to the actual environment, I think we should apply it at 37 ℃, which is the human gut environment.
3. What do you think is the difference between each strain if the genetic consistency of listeria is all greater than 99%?
4. It is necessary to add to the manuscript how much the strains match genetically .
5. This samples should be free of other bacteria to some extent. How did you prevent the removal of bacterial cells and spores from the sample surface?
6. How did you set up each control group?
7. Add to your consideration about the limitations & findings of your research.
Reviewer 3 Report
Dear authors,
The comment on the manuscript is a PDF attachment file, please check it out.
Best regards,
Manuscript ID: foods-2231594-peer-review-v1 The topic “Modelling the growth of six Listeria monocytogenes strains in 2 smoked salmon pâté” is an interesting but not high level of novelty in the field. However, some point of manuscript needs to revise and make it clear for correct understanding. Major revision is required.
Comments are below…
1. Typographical errors exist throughout the manuscript. Rectify carefully.
2. The plagiarisms in the whole manuscript need to revise. The level of similarity seems too high (Almost 40%). The PDF file for similarity checking is attachment file, please check it out.
3. What is the objective for selecting only the smoked salmon? How the fresh one? Please justify.
4. Referred to Figure 1 and 2, It seems no difference was observed in the results of all six L. monocytogenes. So, how to use this point of data for predicting in the next step of trail?
5. The research article address on the application of pathogen strains with the mathematic modeling, if they had to enter the biological systems what will be the ways of elimination or fate of disposal from the biological food chain. Please justify this point.
6. For the introduction section, the author should be added more literature on related previous studies and focus on the vital find out for current work. The vital point or pain point of his study is still not clear. Please consider this one too.
7. All of figure is not clear, try to increase the resolution. Also, the figure legend needs to give the significant detail. 8. In the conclusion seems to be in general and is not given separately, it is highly recommended to include limitation of the study and potential future research goals
Round 2
Reviewer 1 Report
The manuscript can be accepted.
Reviewer 2 Report
Accept (minor edits): The article is acceptable for publication
Reviewer 3 Report
Dear authors,
Thank you for your prompt revision. The current version of a manuscript is fine and it is OK to accept for publication. Congratulation.
Best regards,